# Precision Medicine for Colorectal Cancer with Liquid Biopsy and Immunotherapy

**DOI:** 10.3390/cancers13194803

**Published:** 2021-09-25

**Authors:** Satoshi Nagayama, Siew-Kee Low, Kazuma Kiyotani, Yusuke Nakamura

**Affiliations:** 1Department of Gastroenterological Surgery, Cancer Institute Hospital, Japanese Foundation for Cancer Research, Tokyo 135-8550, Japan; satoshi.nagayama@jfcr.or.jp; 2Department of Surgery, Uji-Tokushukai Medical Center, Kyoto 611-0041, Japan; 3Cancer Precision Medicine Center, Japanese Foundation for Cancer Research, Tokyo 135-8550, Japan; siewkee.low@jfcr.or.jp (S.-K.L.); kazuma.kiyotani@jfcr.or.jp (K.K.)

**Keywords:** liquid biopsy, minimal residual disease, cancer precision medicine, immune checkpoint inhibitor, neoantigen, personalized immunotherapy, neoantigen vaccine

## Abstract

**Simple Summary:**

There are some challenges to improve the clinical outcome of colorectal cancers (CRCs) by implementing new technologies, such as early detection of recurrence/relapse and selection of appropriate drugs based on the genomic profiles of tumors. For example, the genomic characteristics of tumors can be analyzed by blood-based tests, namely ‘liquid biopsies’, which are minimally-invasive and can be performed repeatedly during the treatment course. Hence, liquid biopsies are considered to hold great promise to fill these gaps in clinical routines. In this review, we addressed clinical usefulness of liquid biopsies in the clinical management of CRC patients, including cancer screening, detection of minimal residual disease, selection of appropriate molecular-targeted drugs, monitoring of the treatment responsiveness, and very early detection of recurrence/relapse of the disease. Furthermore, we discussed the possibility of adoptive T cell therapies and a future personalized immunotherapy based on tumor genome information.

**Abstract:**

In the field of colorectal cancer (CRC) treatment, diagnostic modalities and chemotherapy regimens have progressed remarkably in the last two decades. However, it is still difficult to identify minimal residual disease (MRD) necessary for early detection of recurrence/relapse of tumors and to select and provide appropriate drugs timely before a tumor becomes multi-drug-resistant and more aggressive. We consider the leveraging of in-depth genomic profiles of tumors as a significant breakthrough to further improve the overall prognosis of CRC patients. With the recent technological advances in methodologies and bioinformatics, the genomic profiles can be analyzed profoundly without delay by blood-based tests—‘liquid biopsies’. From a clinical point of view, a minimally-invasive liquid biopsy is thought to be a promising method and can be implemented in routine clinical settings in order to meet unmet clinical needs. In this review, we highlighted clinical usefulness of liquid biopsies in the clinical management of CRC patients, including cancer screening, detection of MRD, selection of appropriate molecular-targeted drugs, monitoring of the treatment responsiveness, and very early detection of recurrence/relapse of the disease. In addition, we addressed a possibility of adoptive T cell therapies and a future personalized immunotherapy based on tumor genome information.

## 1. Introduction

In spite of remarkable progress in diagnosis and treatment of colorectal cancer (CRC), there are still big challenges to improve the overall prognosis. Current diagnosis of recurrence/relapse is based on tumor biomarkers or imaging modalities including CT, MR, and PET examinations, which fail to detect minute lesions (micrometastases). It should be favorable to know the precise condition of the disease earlier than imaging diagnosis and initiate a proper treatment before clinically overt recurrences are identified. In addition, most patients who receive systemic chemotherapy become resistant during the treatment course and end up in the termination of the treatment with the standardized guidelines. A suitable drug should be selected and provided based on the molecular biological profiles of the tumors before the disease culminates in a far-advanced stage.

With the advance of genomic sequencing technologies typified by next-generation sequencing (NGS), it becomes much easier, more rapid, and less expensive to access comprehensive genomic information of the tumors. There is a high possibility that usage of the in-depth genomic profiles contributes to detection of recurrent/relapsed tumors and to proper choice of beneficial treatment options, including molecularly targeted therapies. In order to obtain the genomic profiles, blood-based tests, so called ‘liquid biopsies’, are considered to be a more useful method as compared to tumor biopsies, since liquid biopsies can be performed repeatedly and less invasively during a monitoring period and can provide the genomic information that is highly concordant with that of tumor biopsies. Hence, by leveraging the genomic profiles obtained by liquid biopsies, recurrence/relapse can be detected earlier than the current imaging diagnosis, and more suitable therapies can be provided without the delay in the time course of the treatment. Furthermore, analyzing the comprehensive genomic information, which also can be supplemented by liquid biopsies, will lead to a new development of effective immunotherapy related to mutations and/or neoantigens. In this review, we focus on potential roles of liquid biopsy in terms of clinical management of CRCs ranging from early detection of recurrence/relapse to acquisition of definitive clues leading to a promising and beneficial treatment, and touch on the potential of a promising immunotherapy in the treatment of CRCs (Figure 1).

## 2. Cancer Screening Using Liquid Biopsy

CRC is often diagnosed at a late stage owing to the lack of specific symptoms in early stages [1]. It is clinically important to develop an easy, cheap, and sensitive cancer screening method that detects cancer or precancerous lesions before clinical symptoms arise. If we can, we are able to begin treatment at earlier stages and increase probability of cure of the disease.

Colonoscopy screening remains the gold standard for early-stage diagnosis of CRC and has led to the reduction of CRC-related mortality [2,3]. Although colonoscopy is certainly effective, it is an invasive and relatively expensive procedure to screen CRC. Recent advancement of noninvasive screening approaches that were approved by the US Food and Drug Administration (FDA) includes stool-based tests and multitarget stool DNA tests (e.g., Cologuard) summarized in Table 1.

A follow-up diagnostic colonoscopy is performed if abnormalities are indicated by these noninvasive tests. Even though the existing noninvasive stool-based tests for colon cancer have shown high sensitivity and specificity, adherence remains low. An observational study from Germany indicated the improved compliance to CRC screening using these tests. In this study, 97% of the subjects who had refused colonoscopy accepted an alternative noninvasive method; 83% of them chose the blood test, only 15% chose the stool test, and the remaining people refused to receive any screenings [16].

To improve the low adherence of colonoscopy and stool-based tests, liquid biopsy approaches, which have progressed substantially, are likely to be suitable to apply for the screening of CRC using blood samples. In 2016, the FDA approved the first blood-based screening test, Epi proColon, that possibly detects the promoter methylation status of the septin 9 (SEPT9) gene in cell-free DNAs (cfDNAs) for colon cancer using qPCR. Methylated SEPT9 level is increased in CRC and thus serves as a differential biomarker for early detection of CRC. The Epi proColon assay showed sensitivity and specificity values ranging from 68 to 72% and 80 to 82%, respectively (Table 1) [17,18]. Importantly, results from the multicenter randomized ADMIT trial indicated that adherence of Epi proColon blood-based screening was 99.5% compared with 88.1% for the FIT stool-based test, demonstrating a preferential acceptance of the blood test [12].

In addition to methylation signatures, Cohen and colleagues reported a multianalyte detection systems (CancerSEEK) by combining the detection of specific mutations of circulating tumor DNA (ctDNA) with conventional biomarkers for the detection of eight common surgically resectable cancer types. The CancerSEEK approach detected cancer with a sensitivity range from 69% to 98% and a specificity of 99% [13]. Specifically, 65% (252/388) of stage I–III resectable CRC were positive with CancerSEEK [13]. A subsequent prospective interventional study, DETECT-A (Detecting cancers Earlier Through Elective mutation-based blood Collection and Testing) evaluated 10,006 women with no prior history of cancer and followed-up for 12 months with the combination of the CancerSEEK study and imaging. Among those participants, 127 of 134 had a positive blood test underwent PET–CT imaging examination to evaluate the presence or absence of as well as the location of cancer. A total of 26 women were diagnosed to have a cancer, and 65% of them were found to be at a localized stage, potentially amenable to surgical resection [19].

On the other hand, Guardant Health has initiated the ECLIPSE study (https://clinicaltrials.gov/ct2/show/NCT04136002, accessed on 26 April 2021) for early detection of CRC with the LUNAR-2 blood test. The LUNAR-2 assay could detect somatic variants, methylation alterations, and other epigenomic changes and reported a high sensitivity in detecting CRC. This assay will be further tested on approximately 10,000 individuals aged 45–84 who are at average risk for CRC.

GRAIL started the Circulating Cell-free Genome Atlas Study (CCGA) as a discovery study and found that whole-genome bisulfite sequencing (WGBS) interrogating genome-wide methylation patterns outperformed whole-genome sequencing (WGS) and targeted sequencing approaches interrogating copy-number variants (CNVs) and single-nucleotide variants (SNVs)/small insertions and deletions, respectively [20,21]. GRAIL established a high-specificity (low false positive rate) targeted bisulfite sequencing, which focused on more than 100,000 methylation sites in our genome and assessed methylation patterns to evaluate the presence or absence of cancer with machine learning. The results of CCGA and the STRIVE study reported a sensitivity of 67.3% for 12 cancer types at stages I to III with an accuracy of 93% to predict tissue of origin [14].

Cristiano et al. [15] developed a method called DELFI (DNA evaluation of fragments for early interception) for early cancer detection. This method utilized the differences of genome-wide cfDNA fragmentation profiles as well as machine learning to distinguish cancer patients from healthy individuals. DELFI detected 152 of 208 patients with eight cancer types including breast, CRC, lung cancer, ovarian cancer, pancreatic cancer, gastric cancer, and cholangiocarcinoma. The overall sensitivity and specificity were 73% and 98%, respectively and 81% and 95% for CRC, respectively. Furthermore, among the 126 patients who were evaluated by both targeted sequencing and DELFI, the sensitivity of DELFI alone was 66% (83 of the 126 patients), but when combining both tests, the sensitivity improved to 82% (103 of the 126 cases) [15].

## 3. Genomic Analysis for Selection of Molecular-Targeted Drugs

In cases of systemic/distant recurrences after the curative resection of primary CRCs or in those with surgically unresectable stage IV CRCs, intensive systemic chemotherapy is provided to halt the progression of the disease. However, it is difficult to completely eradicate cancer cells using the current regimens of systemic chemotherapy, so novel therapies based on the genomic profiles of the tumors of individual patients should be developed. Some of targeting genetic mutations include KRAS, BRAF, HER2, and microsatellite instability (MSI), which are leveraged in the current clinical setting [22] (Table 2).

As for application of the KRAS mutation status to select chemotherapy regimens, phase III clinical trials such as CRYATAL, OPUS, CO.17, and FIRE-3 have shown that the benefit of adding cetuximab (anti-epidermal growth factor (EGFR) antibody) to FOLFOX or FOLFIRI was confined to patients with CRCs not having KRAS mutations [29,30,31,32,33,34,35]. With respect to KRAS-mutant tumors, the complexity of the signaling network of the KRAS-mutant alleles has made it difficult to develop molecularly-targeted therapies against KRAS mutations. Mutant KRAS protein has thus been regarded as an undruggable target, so most therapeutic strategies have been designed to inhibit downstream effector pathways such as the ERK/MAPK cascade. However, the clinical efficacy of targeting downstream effectors has been marginal [36]. Two covalently-binding inhibitors, AMG510 (Sotorasib) and MRTX849 (Adagrasib), which specifically target the KRAS G12C mutation, have recently been developed [37,38,39,40,41,42], and their encouraging efficacy in solid tumors harboring the KRAS G12C mutation including non-small cell lung cancers (NSCLCs) and CRCs has been demonstrated in several clinical trials [43,44,45,46,47]. However, most NSCLC patients with the KRAS G12C mutant showed a favorable response to selective KRAS G12C inhibition, while CRC patients harboring the same mutation rarely revealed clinical benefits. This drug resistance is speculated to result from a possible mechanism where a novel mutation can appear [48,49] and/or the feedback reactivation of the RAS pathway following KRAS G12C inhibition may occur. To overcome the acquired resistance by the adaptive RAS pathway feedback reactivation in CRCs, combinatorial targeting of EGFR and KRAS G12C or, theoretically, concomitant inhibition of SHP2 and KRAS G12C is expected as a promising treatment strategy, since SHP2 mediates signaling from multiple receptor tyrosine kinases to RAS, and its inhibition can more comprehensively hamper the feedback reactivation [50,51].

Regarding the treatment for CRCs with the BRAF V600E mutation, the administration of a BRAF inhibitor (vemurafenib) alone showed only limited clinical efficacy compared to the favorable responses observed in melanoma patients [52,53]. As in the case of KRAS G12C inhibition, adaptive feedback reactivation of the RAS-signaling pathway is considered to be a major mechanism of therapeutic resistance or poor response. Specifically, BRAF inhibition in cancers with the BRAF V600E mutation led to loss of negative feedback signals through the MAPK pathway in CRCs, resulting in receptor tyrosine kinase-mediated reactivation of MAPK signaling by wild-type RAS and RAF [54,55,56,57,58]. The concomitant administration of dabrafenib and trametinib therefore has a substantial impact on clinical efficacy in a subset of patients with BRAF-V600E CRCs [59]. Furthermore, combined BRAF + EGFR + MEK inhibitions are tolerable and result in favorable clinical responses and a significantly longer overall survival compared to standard therapy in patients with BRAF-V600E CRCs [60,61].

Regarding HER2-positive CRCs, HER2-inhibiting antibodies and small molecules can suppress the activity of HER2-amplification or mutations [62]. According to several clinical trials including HERACLES, MyPathway, and DESTINY-CRC01, HER2-targeted therapies including anti-HER2 antibody conjugated with or without cytotoxic drugs showed promising and long-lasting outcome in HER2-positive CRCs that had been refractory to standard treatment [63,64,65,66,67]. Although HER2 amplification is identified in only 2–3% of CRCs, these results could provide hope to a substantial number of CRC patients who have experienced progression of the disease with the standardized guidelines.

In MSI-high CRCs due to mismatch-repair deficiency, immune checkpoint inhibitors such as pembrolizumab and nivolumab have shown a significant clinical benefit in clinical trials including CheckMate-142 and KEYNOTE-164 [68,69,70,71,72,73]. Furthermore, pembrolizumab monotherapy has led to clinically meaningful improvements in health-related quality of life compared with chemotherapy in MSI-high CRC patients (KEYNOTE-177) [74,75]. The administration of immune checkpoint inhibitors is thus regarded as a first-line treatment option for this population [76]. However, acquired resistance to anti-PD-1 immunotherapy was reported in a subset of cases where the expression of MHC and/or B2M was reduced or either of these genes were lost in tumor cells, leading to impaired antigen presentation and resulting in immune evasion [77,78,79]. Hence, therapies targeting CTLA4 or PD-1 still have some limitations in treatment efficacy. It is also an intriguing approach to upregulate MHC-I expression to enhance sensitivity to immunotherapy [80].

NGS-based targeted-gene panel tests have recently been used in a clinical setting to identify patients with actionable genetic alterations for enrollment in genotype-matched clinical trials. According to the mutational landscape of metastatic cancers of more than 10,000 patients with clinical sequencing using a comprehensive assay MSK-IMPACT, one or more potentially actionable genetic alterations was detected in 36.7% of the patients, and 11% could be enrolled to genome-guided clinical trials [81]. In the literature covering genomic testing of advanced cancers, a small proportion of patients (4–31%) had a chance to receive genetic-alteration-matched therapy [82,83,84,85,86,87,88,89,90,91,92,93,94]. Furthermore, an observational study involving more than 1000 patients showed that overall response rates, time-to-treatment failure, and overall survival were higher with matched targeted therapy than those observed without matching, suggesting that identifying specific genetic alterations and choosing therapy based on these alterations are associated with a better prognosis than standard systemic therapy [83]. This finding is consistent with a recent meta-analysis of phase I trials that showed a higher overall response rate (30.6% vs. 4.9%, *p* < 0.001) and median progression-free survival (5.7 months vs. 2.95 months, *p* < 0.001) for genotype-matched trials, compared with non-selected therapies [95].

With the advent of WGS and whole-exome sequencing (WES), we can share more comprehensive information on the genomic alterations of individual tumors than with targeted gene-panel sequencing [22]. A recent WGS analysis comprising 2520 samples in 22 types of metastatic tumors showed that 62% of these tumors harbored at least one actionable mutation [96]. With the advances in sequencing technologies, more actionable biomarkers and/or oncogenic mutations have been detected in individual cancers. While it might suggest that a high proportion of actionable alterations are detectable in cancer patients by WGS and WES, the clinical benefits for cancer patients are still very limited. The limited contribution of gene panel tests is attributable to a variety of reasons ranging from patient-dependent factors such as health deterioration in those with an advanced cancer and patient preferences to physician-dependent factors including strict inclusion criteria of clinical trials, the cost of off-label use of drugs, and limited supply of molecular-targeted drugs [84,85,88]. Since genomic profiles can alter clinical management in diverse scenarios, the combination of comprehensive molecular testing and better access to genome-guided trials can improve rates of clinical trial enrollment, thereby enabling precision cancer medicine on a large scale. To expand opportunities for genome-matched therapies, further development of novel molecular-targeted drugs and other treatment options is urgently required.

## 4. Detection of Minimal Residual Disease (MRD) by Liquid Biopsy after the Curative Resection

Liquid biopsy has been introduced as a new diagnostic test based on the genomic or proteomic analysis of circulating tumor cells or tumor-derived components such as ctDNA, microRNAs, long non-coding RNAs, proteins/peptides, and extracellular vesicles from peripheral blood or other body fluids. The liquid biopsy is a minimally-invasive and repeatable technique that could play an essential role in screening and diagnosis, and detect relapse/recurrence prior to detection of imaging tests in cancer patients [97]. Recent studies have provided strong evidence that the results of mutational analysis with liquid biopsies are highly concordant with those of tumor DNAs (97–100%) across multiple cancer types and that they can be used reliably to match patients to mutation-directed clinical trials [98,99,100,101,102,103,104,105]. It is of particular note that highly sensitive liquid biopsy assays can now be applied to detect and characterize minimal residual disease (MRD), which can be defined as persisting cancer cells disseminated from the primary lesion to distant organs or into blood circulation in patients who have no radiological signs of relapse/recurrence after the curable resection of primary (and metastatic) tumors. Since MRD is considered as an occult stage of cancer progression, minimally-invasive liquid biopsies are useful for the sensitive monitoring and early detection of the disease. In addition, liquid biopsies can obtain information on the molecular evolution of MRD during tumor progression, which provides insights into therapeutic targets and resistance mechanisms relevant to the clinical management of cancer patients. Therefore, further characterizing the biology of MRD through liquid biopsies can lead to the development of therapy to delay or even prevent clinically-overt metastasis [106,107].

Because of the lesser invasiveness of liquid biopsies, monitoring of blood samples collected at primary diagnosis and at subsequent time points during the follow-up period after surgery is feasible in the clinical setting. Liquid biopsy-based post-operative assessments would be able to detect tumor relapse/recurrence a few to several months earlier than current radiological imaging modalities [101,108,109,110]. According to a recent study, patients with positive ctDNA after their surgery showed a significantly shorter recurrence-free survival than patients without ctDNA [111]. This implies that screening of ctDNA with liquid biopsy can be useful in identifying patients at high risk of post-operative recurrence, and that serial screening of ctDNA would allow early detection of tumor relapse/recurrence.

NGS-based ultra-deep sequencing of cfDNA and/or epigenetic analysis such as DNA methylation-based liquid biopsy that enable us to reliably detect minute amounts of ctDNA might complement current imaging procedures for post-surgical surveillance of recurrences [14,112,113,114,115,116,117,118,119,120]. Earlier detection of the presence of small numbers of tumor cells with liquid biopsy should be beneficial as it could potentially lead to alternative interventions with new types of post-adjuvant therapies before overt metastasis is identified [121]. Furthermore, liquid biopsies offer key advantages such as the capability to more rapidly identify targetable alterations, thereby facilitating genotype-guided clinical-trial enrollment with a more rapid turnaround time [103]. It will thus be feasible in the future to stratify CRC patients and to choose the most appropriate therapy based on real-time genetic information through liquid biopsies as a kind of personalized medicine.

As mentioned above, MRD surveillance is a viable step for improving the post-operative prognosis. Liquid biopsy technologies can be used to identify patients who have a high risk of disease recurrence following curative resection of primary tumors. Furthermore, liquid biopsy-based analysis will be essential in developing reliable surrogate biomarkers of relapse in patients without imaging-detectable metastatic lesions, possibly providing a better chance of cure of the disease. In the treatment for breast cancers, the use of liquid biopsies for the sensitive and specific detection of tumor cells that cannot be detected by the most sensitive contemporary imaging modalities would enable testing of new adjuvant or post-adjuvant treatment strategies to prevent progression to imaging-detectable metastases [121].

Genomic profiles of ctDNA in individual patients will provide unique information that might indicate genes involved in cancer dormancy or the progression from MRD to clinically judged metastasis at present [122]. Mutations conferring sensitivity or resistance to targeted therapies can also be monitored by ctDNA assessment. Identification of these biomarkers by liquid biopsy could help physicians in fine-tuning of the treatment regimen and/or treatment period to optimize the benefit of the treatment, since the sensitivity or resistance of tumor cells to available therapies could be indicated by liquid biopsy-based analysis. However, for accurate disease monitoring by liquid biopsy, false-positive findings should be taken into consideration when the concentration of ctDNA is very low and results are influenced by clonal hematopoiesis or possible benign lesions of another origin [123,124,125,126,127]. In terms of surveillance of MRD, longitudinal ctDNA analyses might provide intriguing findings for tumor evolution [128]. Thus, liquid biopsies can enhance our understanding of the evolution and eventual outgrowth of MRD. In addition, serial profiling of the ctDNA genome following each line of treatment could predict the treatment efficacy and acquired resistance to chemotherapy, thus providing accurate prediction of prognosis. Taken together, blood-based liquid biopsy analyses have the potential to cause a paradigm shift in the diagnostic and therapeutic fields in the MRD context, thereby accelerating precision medicine for CRC patients.

## 5. Estimation of Treatment Responsiveness, Detection of Acquired Mutations, and Early Detection of Disease Progression with Liquid Biopsy

Standard post-surgical and post-treatment surveillance include radiologic imaging, colonoscopy, and serum biomarkers, including carcinoembryonic antigen (CEA) and carbohydrate antigen 19-9 (CA19-9). Although CEA and CA19-9 assessments are convenient and cost-effective, these biomarkers have relatively low sensitivity and specificity, and their clinical value in evaluating CRC recurrence is not sufficient [129,130]. Evaluation of responses to therapies in oncology is mainly based on Response Evaluation Criteria In Solid Tumors (RECIST) criteria by evaluating changes of morphological (CT or MRI) or metabolic (18FDG-PET/CT) activity of target neoplastic lesions. Nevertheless, these conventional approaches may not be able to detect minimum tumor lesions and thus have difficulty providing real-time assessment of drug-resistant tumor cells that cause disease progression. Contrarily, liquid biopsy emerged as a complimentary assay to provide real-time assessment of tumor’s molecular profiles. Liquid biopsy allows the evaluation of the clonal evolution of tumor during the course of treatment and early detection of treatment-resistant tumor cells and detects disease progression much earlier than conventional radiological procedures [108,131,132]. Based on a report that conducted serial surveillance of ctDNA profiles of 130 stage I to III CRC revealed disease recurrence up to 16.5 months in advance of standard-of-care radiologic imaging and CEA-based surveillance (mean, 8.7 months; range, 0.8–16.5 months) [108].

Tie and colleagues conducted a well-designed prospective study to evaluate ctDNA profiling at pretreatment, post-chemoradiotherapy, and 4–10 weeks after surgery in 159 locally advanced rectal cancer [133]. Patients, who were ctDNA-positive at post-operation and were indicative of residual disease, resulted in high probability of recurrence. Importantly, significantly-improved overall survival was observed among the ctDNA-positive patients who received adjuvant systemic chemotherapy, compared with those who did not receive chemotherapy (chemotherapy: HR 10.0; *p* < 0.001; without chemotherapy: HR 22.0; *p* < 0.001) [133]. Another prospective study evaluated the early changes of ctDNA levels among 53 metastatic CRC patients receiving standard first-line chemotherapy. Significant reductions in ctDNA levels (median 5.7-fold; *p* < 0.001) between the pre-treatment and pre-cycle 2 were observed. The ctDNA changes in these patients were correlated with measurement with CT imaging at 8–10 weeks of the treatment. It is notable that patients with significant decrease (less than one-tenth) in ctDNA molecules from pre-treatment to pre-cycle 2 showed a tendency of longer progression-free survival compared with those showing lesser decreases (median 14.7 versus 8.1 months; HR = 1.87; *p* = 0.266) [134]. These results indicated serial assessment of ctDNA profiles could be used as a marker to evaluate the treatment efficacy by characterizing patients who might have a benefit from chemotherapy. Based on these observational studies, several interventional clinical trials were initiated to confirm the clinical utility of ctDNA profiling in CRC. The DYNAMIC II trial (https://www.anzctr.org.au/Trial/Registration/TrialReview.aspx?ACTRN=12615000381583, accessed on 26 April 2021) focused on stage II colon cancer patients who have positive ctDNA and will receive adjuvant chemotherapy and those having no detectable ctDNA and will not receive adjuvant therapy. The primary objective of DYNAMIC II is to determine if ctDNA information of postoperative tests would affect the number of patients receiving adjuvant therapy and recurrence free survival. The DYNAMIC III study (https://www.anzctr.org.au/Trial/Registration/TrialReview.aspx?id=373948, accessed on 26 April 2021) recruited stage III colon cancer patients to investigate the utilization of ctDNA state in the postoperative liquid biopsy test for dosage adjustment and examine the outcome with recurrence-free survival.

Targeted inhibitors such as panitumumab and cetuximab are routinely used as one of the combination regimens with standard chemotherapy for CRC patients. However, a majority of these patients inevitably develops resistance to these treatments. Understanding the resistance mechanisms simultaneously with a surveillance system is essential to improve the management quality of CRC patients. With the high overall concordance between results of liquid biopsy ctDNA and tumor biopsy, ctDNA could be used as a surrogate marker for real-time assessment of the disease status during the treatment course [135,136]. For instance, it has been repeatedly reported that metastatic CRC patients who harbor RAS mutations responded poorly to anti-EGFR treatment [137,138]. ctDNA could be used as a surrogate diagnostic test for CRC patients to select or not to select anti-EGFR treatment. The strategy of the PROSPECT-C phase II CRC clinical trial was to evaluate cetuximab treatment efficacy in RAS wild-type patients by combining serial cfDNA profiling and matched sequential tissue biopsies with imaging and mathematical modeling of cancer evolution. As a result, liquid biopsies were managed to capture spatial and temporal heterogeneity that might affect the resistance to anti-EGFR antibodies [139]. It is noteworthy that the primary or acquired resistance to EGFR blockade was related to alterations in genes KRAS, NRAS, MET, HER2, FLT3, EGFR, and MAP2K1 using ctDNA profiling [140]. In the HERACLES, a phase II trial aimed at testing trastuzumab and lapatinib in metastatic CRC patients with HER2-amplified CRC, HER2 copy number alteration (CNA) was confirmed in ctDNA in 51 of the 52 plasma samples. In addition, ctDNA analysis identified gene alterations that could be associated with resistance (HER2, RAS, PIK3CA mutations) in the majority of (>85%) refractory patients to HER2 blockade. Interestingly enough, through longitudinal monitoring of individual liver metastasis in ctDNA, heterogeneous patterns of radiographic response to treatments could be interpreted properly in association with clinical RECIST assessment using liquid biopsy-derived genomic information [141]. Taken together, liquid biopsy ctDNA showed superiority to tissue biopsy in detecting genetic alterations that cause treatment resistance.

## 6. Personalized Immunotherapy Based on Genome Information of Tumors

It has been reported that anti-tumor T cells infiltrated tumor sites to eliminate cancer cells, and tumors with higher CD8+ T cell infiltration showed better clinical outcomes [142]. These tumor-infiltrating anti-tumor T cells are known to recognize cancer-specific antigens, including shared antigens and neoantigens that were generated by epigenetic alterations or somatic mutations that occurred in cancer cells, and to play an important role in anti-tumor immunity. The numbers of somatic mutations, which is referred to as tumor mutation burden (TMB), were significantly associated with clinical outcome of CRC patients receiving immune checkpoint inhibitors; patients with MSI-high- or polymerase ε-mutated (POLE)-type CRC showed significantly better responses to immune checkpoint inhibitors. It is now well known that these CRC patients have hundreds of genetic alterations causing amino acid substitutions that generate tumor-specific neoantigens. On the contrary, the response rate to immune checkpoint inhibitors in patients with microsatellite-stable (MSS) CRC was as low as less than 10% because of low numbers of neoantigens and low levels of T cell infiltration into tumor sites. Therefore, novel approaches to enhance cytotoxic T lymphocyte (CTL)-mediated anti-tumor immune responses in cancer immunotherapies are highly expected. Recently, TMB measured in the blood using ctDNA were reported to be well correlated with TMB in cancer tissues [143]; therefore, TMB in blood may also be a useful predictive biomarker of the clinical benefit of ICIs. Many combination therapies with immunotherapy have been investigated, including radiotherapy, chemotherapy, and molecular-targeting drugs [144]. Radiation induces immunogenic tumor cell death and possibly contributes to enhancement of T cell immune responses, leading to systemic tumor control through the abscopal effect [145,146,147]. Adoptive transfer of tumor-infiltrating T-lymphocytes (TILs) based on the use of T cells that have infiltrated into a patient’s tumor is also one of the approaches to enhance CTL-mediated anti-tumor immune responses. Adoptive TIL therapy was extensively investigated in melanoma and was shown to be effective at 50% or higher objective response rate [148,149]. The efficacy of adoptive TIL therapy has also been assessed in several clinical trials for solid tumors, including CRC [150,151,152,153]; however, the response rates were lower than those reported in melanoma, possibly because of the lower presence of tumor-reactive CD8+ T cells in the tumor microenvironment because the number of somatic mutations generating neoantigens was much lower in most sporadic CRCs [154].

Marits et al. investigated the immunological role of sentinel lymph nodes and conducted a pilot study of adoptive T cell therapy using in vitro expanded autologous lymphocytes isolated from sentinel lymph nodes [155,156]. In this study, four of nine stage IV CRC patients, who received this adoptive immunotherapy, showed complete tumor regression with median survival of 2.6 years as compared with 0.8 years in controls who received the standard therapy. Zhen et al. also reported the possibility of adoptive T cell therapy using T cells in lymph nodes [157], in which the adoptive T cell therapy group showed a significantly better 2-year survival rate of 55.6% than the 17.5% of the control group. We recently characterized the T cell receptor (TCR) repertoire of a total of 203 regional lymph nodes from 23 CRC patients and compared those in primary cancer tissues [158]. In the study, we found that regional lymph nodes, especially metastasis-positive lymph nodes, contained T cells with TCR shared with primary cancer tissues. These data suggest that lymph nodes might be a better source of T cells for adoptive T cell therapy.

In the analysis of TILs in CRC patients, it has been reported that neoantigen-reactive T cells are present in TIL populations, and adoptive transfer of the expanded TILs showed tumor regression [150,159,160]. Interestingly, one of the target neoantigens is the peptide corresponding to a KRAS G12D mutation, which was presented on HLA-B08:02 [150,159,161]. Since KRAS mutation was observed in 40–50% of CRCs, KRAS mutations may be good targets of neoantigen-based cancer vaccine or TCR-engineered T cell therapies.

## 7. Conclusions

Blood-based liquid biopsy represented by ctDNA analysis is a very promising tool to play a critical role in several aspects in the clinical management of CRC patients. With the accumulation of the relevant data, liquid biopsy is expected to become an indispensable and routine method to monitor the disease status quickly and precisely for matching the most beneficial therapy in the near-future clinical setting, thereby improving the treatment outcome of CRCs. In addition, genome-guided novel immunotherapy might further improve the prognosis of CRC patients. Current neoantigen-based personalized immunotherapy needs tissue samples or biopsies to obtain accurate information of somatic genomic alterations in individual cancer patients. However, it is sometimes difficult to obtain a large enough amount of tumor tissues; therefore, improvement of ctDNA analysis [162], including WGS/WES of cfDNA, could be important to expand neoantigen-based therapies, although it is still challenging.

## Figures and Tables

**Figure 1 cancers-13-04803-f001:**
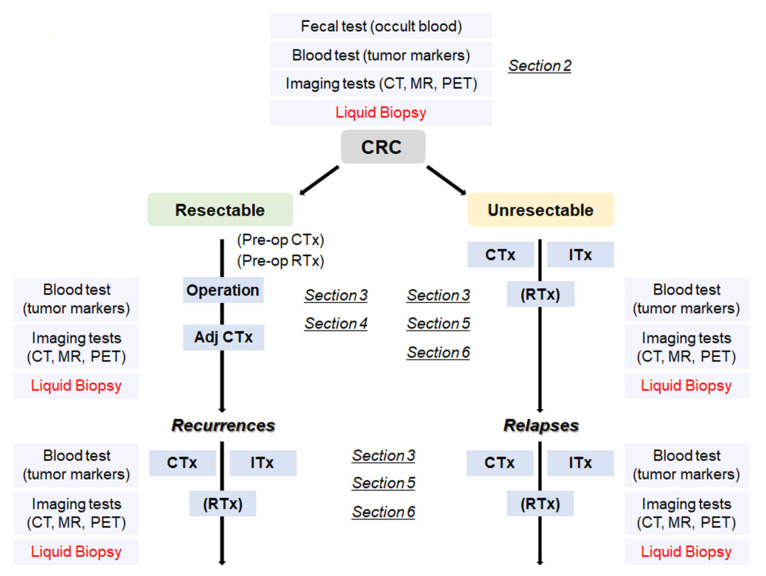
A flowchart from the diagnosis of CRC to treatment according to the disease progression. CRC: colorectal cancer, Adj CTx: adjuvant chemotherapy, CTx: chemotherapy, RTx: radiotherapy, Itx: immunotherapy.

**Table 1 cancers-13-04803-t001:** Up-to-date stool- and blood-based screening method.

Screening Method	Type of Sample	Description	Overall Performance
High sensitivity Guaiac-based fecal occult blood test (gFOBT) -FDA approved	Stool	Check for hidden blood in stool using chemical guaiac.	In randomized controlled trials, screening with FOBT reduced CRC mortality rates by 15% to 33% [4,5,6] Specificity: 86.7% to 97.7%; Sensitivity: 33.3% to 79.4% [7,8]
Fecal Immunochemical test (FIT) -FDA approved	Stool	Utilization of antibody specific to human globin to directly detect hemolyzed blood from the stool.	Randomized control: On-going Specificity: 81% to 96%; Sensitivity: 65% to 95% [9,10]
Multitarget stool DNA test (Cologuard) -FDA approved	Stool	Detect altered DNA for cancer in cells shed from the lining of the colon and rectum into the stool. Two highly discriminant methylated genes (*BMP3* and *NDRG4*), 7 most informative point mutations on *KRAS*, a marker for total human DNA (*β-actin*), and fecal hemoglobin.	Specificity: 86.6%; Sensitivity: 92.3% [11]
Epi proColon -FDA approved	Blood	Detect promoter methylation status of septin 9 (*SEPT9*) gene in cell free DNA for colon cancer	Specificity: 80 to 82%; Sensitivity: 68 to 72% [12]
CancerSEEK	Blood	Detect specific mutations of ctDNA with conventional peptide biomarkers	All 8 cancer types including CRC Specificity > 99% Sensitivity: 33% to 70% [13]
GRAIL (CCGA and STRIVE)	Blood	Targeted bisulfite sequencing focus in more than 100,000 methylation regions from the genome. Assess methylation patterns to evaluate the presence or absence of cancer with the aid of machine learning.	All 12 cancer types including CRC Specificity: 98.3% to 99.8% Stage I–III sensitivity: 60.7% to 73.3% [14]
DELFI	Blood	Genome-wide cfDNA fragmentation profiles and uses machine learning to distinguish between cancer patients and healthy individuals.	Specificity: 95%; Sensitivity: 81% [15]

**Table 2 cancers-13-04803-t002:** Genomic biomarkers in CRCs.

Gene	Biomarkers	Frequencies (%)	Anticancer Agents
*KRAS*	Wild type	60 [23]	Cetuximab Panitumumab
*KRAS*	G12C	8 [23]	Sotorasib (AMG510) Adagrasib (MRTX849)
*BRAF*	V600E	10 [24]	Vemurafenib Dabrafenib Ecorafenib
*HER2*	Amplification	2–3 [25,26]	Pertuzumab Trastuzumab Lapatinib
*MLH1* *MSH2* *MSH6* *PMS2*	MSI-H	10–15 [27,28]	Nivolumab Pembrolizumab Ipilimumab

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
