# Peer review of "Precision Medicine for Colorectal Cancer with Liquid Biopsy and Immunotherapy"

_cancers, 2021, doi:10.3390/cancers13194803_

Round 1

Reviewer 1 Report

The authors presented a very broad and interesting compendium of information regarding the use of liquid biopsies and precision medicine in CRC management. 

I believe the work is sound and should be accepted to publication. In order to improve it further, I'd suggest the authors provide a comparison among the different methods listed in section 2, as well as a comparison between blood and stool samples, to help guide researchers using this information. It would also be interesting to discuss why the genes chosen to be monitored in stool differ from those used in blood based exams. 

Reviewer 2 Report

The manuscript is well written and comprehensively illustrates the
clinical management of CRC patients and the prospects for improving it.
The references used are very well chosen and explained. In summary, the
Review identifies the imminent future in the analysis of liquid biopsy
on which important information can be obtained on the most correct
targeted molecular therapies to be administered to the patient in
association with chemo and radiotherapy. The analysis of ctDNA from
liquid biopsy will allow to identify the prognosis with greater
precision and to personalize the therapy for individual patients on the
basis of the expected response to drugs in the presence of the various
possible genetic alterations. It is also emphasized that this system
constitutes a particular advantage in the event of the development of
resistances in the perspective that the detection and analysis systems
of the ctDNA turn out to be increasingly better and more sensitive. The
sensitivity of the method has and, increased, could have an even greater
impact on the identification of minimal residual disease and its
characteristics. My only advice is to add, where possible, tables or
flowcharts representative of the different points covered in the
paragraphs to focus more immediately on the information you are reading. 
